# *Vernonanthura tweediana* (Baker) H. Rob. (Asteraceae), an Ordinary Bush or an Anti-Inflammatory and Immunomodulator Aromatic Species?

**DOI:** 10.3390/ph17111492

**Published:** 2024-11-07

**Authors:** Isabel D. Machado, Pâmela P. Borges, Larissa J. A. Giacomozzi, Larissa Benvenutti, José R. Santin, Sarah C. dos Santos, Martinho Rau, Iêda M. Begnini, Ricardo A. Rebelo, Henrique D. M. Coutinho, Clara M. G. Lima, Talha B. Emran

**Affiliations:** 1Postgraduate Program in Biodiversity, Regional University of Blumenau, Rua Antônio da Veiga, 140, Itoupava Seca, Blumenau 89030-903, Brazil; isabelm@furb.br (I.D.M.); ppborges@furb.br (P.P.B.); 2Laboratory of Synthesis and Technology, T-313, Regional University of Blumenau, Blumenau 89030-903, Brazil; ljagiacomozzi@furb.br; 3Postgraduate Program in Pharmaceutical Sciences, University of Vale do Itajaí-UNIVALI, Itajaí 88302-901, Brazil; larabenv@gmail.com (L.B.); jrs.santin@univali.br (J.R.S.); 4Postgraduate Program in Chemistry, Regional University of Blumenau, Blumenau 89030-903, Brazil; sarahcastro.chemistry@gmail.com (S.C.d.S.); mrau@furb.br (M.R.); ieda@furb.br (I.M.B.); 5Department of Biological Chemistry, Cariri Regional University, Crato 63105-000, Brazil; hdmcoutinho@gmail.com; 6Department of Food Science, Federal University of Lavras, Lavras 37200-900, Brazil; claramarianalima@gmail.com; 7Department of Pathology and Laboratory Medicine, Warren Alpert Medical School, Brown University, Providence, RI 02912, USA; talha_bin_emran@brown.edu; 8Legorreta Cancer Center, Brown University, Providence, RI 02912, USA; 9Department of Pharmacy, Faculty of Allied Health Sciences, Daffodil International University, Dhaka 1207, Bangladesh

**Keywords:** hydrodistillation, essential oil, chromatographic characterization, caryophyllenes, leukocytes and cytokines

## Abstract

**Background:** Oral traditional knowledge disseminates the use of young leaves of *Vernonanthura tweediana* (Asteraceae) as poultice in feverish cases among family farmers in Santa Catarina, Brazil, but there is a lack of phytochemistry and pharmacological studies on this species, especially those related to volatile organic compounds. **Methods:** In the present study, the dry leaves of *V. tweediana* were subjected to hydrodistillation for the extraction of essential oil. Sample characterization was conducted using GC-FID and GC-MS. To evaluate anti-inflammatory and immunomodulatory activities, the air pouch method was used, and with the exudate obtained, total and differential leukocyte counts were performed, as well as the quantification of total proteins, interleukin (IL) 1β, IL-6, and tumor necrosis factor (TNF). **Results:** The essential oil yield extraction was 0.13%, and sesquiterpenes were the main constituents: α-copaene (5.1%), β-caryophyllene (27.77%), α-caryophyllene (15.1%), germacrene D (14.0%), and bicyclogermacrene (13.3%). A sample was subjected to biological tests, with a reduction in leukocyte migration to the inflammatory focus. The immunomodulatory activity was also pronounced, as the oil modulated all cytokine secretions measured when compared to the control. The high concentration of caryophyllenes might explain these findings. **Conclusions:** Future studies will consider the fractionation of the essential oil to establish the degree of synergism between the constituents.

## 1. Introduction

The Asteraceae family encompasses many species, with several examples within the genus *Vernonanthura* with proven therapeutic activity. Among them, the following members and properties can be highlighted: *V. colorata*, antibacterial and anti-inflammatory [1], *V. cinerea*, analgesic, antipyretic, anti-inflammatory, and cytotoxic [2], *V. phosphorica*, antimicrobial, leishmanicidal, antinociceptive, and anti-inflammatory [3], and *V. montevidensis*, moderate antimycobacterial [4].

*V. tweediana*, popularly known as “chamarritão” or “figatil”, is used by different populations to treat stomach and liver pain [5]. Also known as “assa-peixe”, “chamarrita”, and “mata-pasto”, it is popularly used by different communities for treating respiratory problems [6]. There are reports on the biological activities of *V. tweediana* extracts, such as antibacterial [7], antifungal [8], antioxidant [9], immunomodulatory in mice [10], antinociceptive, and anti-edematogenic [11]; in addition, it presents weak toxicity against *Artemisa salina* and allelopathic activity [12]. Among family farmers in the interior community of the Itajaí Valley of Santa Catarina, southern Brazil, young fresh leaves were used as poultice in feverish cases (RAR, personal communication, 19 October 2019).

The treatment of inflammatory conditions is based on steroidal and non-steroidal anti-inflammatory drugs; however, prolonged use of these medications can cause harmful effects, such as gastrointestinal, cardiovascular, and hepatic disorders. Prospecting new bioactive natural products and extracts, especially those capable of acting in multiple biological pathways, is very attractive, and essential oils commonly composed of several chemical structures may well fit in the search for multitargeted therapy [13]. The anti-inflammatory and immunomodulatory effects of essential oils and their components are known for their potential to reduce oxidative stress through the scavenging of reactive oxygen species (ROS), reducing immune cell activation, and modulating pro-inflammatory responses. Furthermore, essential oils influence the expression of inflammatory mediators such as inducible nitric oxide synthase (iNOS) and cyclooxygenase-2 (COX-2), which play a role in producing gaseous and eicosanoid pro-inflammatory molecules, respectively. Additionally, research evidence indicates that essential oils can inhibit the activation of NF-κB and MAPKs, resulting in a reduction in pro-inflammatory cytokines and chemokines. These targets are essential for regulating the inflammatory process as they operate across various pathways involved in cellular and molecular signaling [14,15].

Given the lack of studies related to the volatile constituents of this species, this work investigates the chemical composition of the essential oil from *V. tweediana* leaves and its anti-inflammatory and immunomodulatory activity, aiming to support its use in folk medicine. In addition, it will provide the scientific basis for the potential use of its essential oil in pharmaceutical formulations, resulting in the valorization of Atlantic Forest biodiversity and stimulating research on the development of phytotherapics.

Biological assays demonstrated notable anti-inflammatory and immunomodulatory activities, including reduced leukocyte migration and modulated cytokine secretion. These effects are likely linked to the high caryophyllene content.

## 2. Results

The investigation conducted is presented in two main headings, extraction of the essential oil of *V. tweediana* and its chemical characterization and the biological assays associated with the air pouch model.

### 2.1. Extraction and Chemical Characterization of Essential Oil from V. tweediana 

To extract the essential oil (EO), the hydrodistillation technique was used, which consists of heating the plant material in direct contact with distilled water. The hydrodistillation product was isolated using liquid–liquid extraction with bidistilled dichloromethane as an extracting solvent. The removal of the solvent and traces of water gave pure essential oil in 0.13% yield.

The sample was analyzed via gas chromatography using a flame ionization detector (GC-FID) and a mass detector (GC-MS). These techniques in association with the arithmetic retention index (AI) are very efficient for the characterization of organic volatile compounds. The corresponding GC-FID and GC-MS chromatograms are given as Appendix A). Although mass spectrometry can detect very low analyte concentrations in a complex mixture such as essential oils, its use for quantification is compromised because ionization efficiencies are variable among other factors. GC-FID gives a better correlation between a peak area of a given constituent and its concentration; furthermore, outstanding reproducibility can be achieved. This can be confirmed by the standard deviation observed in Table 1.

Figure 1 shows a selected expanded chromatogram (18–36 min) and peaks are assigned with the entries 14–35. The peaks 15, 18, 19, 21, and 23 correspond to the major constituents. Expanded GC-MS chromatograms covering entries 1–36 are given as SM (Appendix A).

Based on the chromatogram profile, expressed in the retention times observed in association with the selected analytical conditions [16], it is possible to assert that the essential oil is poor in hydrocarbon and oxygenated monoterpenes since most of them eluted below 18 min. Above 18 min are the hydrocarbon and oxygenated sesquiterpenes, along with the majority of the arylpropanoids. This was confirmed through GC-MS analysis and AIs.

For the identification of each constituent, mass spectra and arithmetic retention indices (AIs) were considered. Table 1 shows the identified constituents with their corresponding AI and concentration. The MS spectra of the constituents and the MS spectra of the identified compound from the NIST library are given as SM (MS Data).

Hydrocarbon sesquiterpene was the main class of secondary metabolites among the 36 compounds characterized, with safrole as the only representative of arylpropanoids but in very low concentration. The caryophyllenes (α and β) comprise approximately 43% of the sample and were the two major constituents, followed by germacrene D (14%) and bicyclogermacrene (13%). Spathulenol and caryophyllene oxide were the main bioactive oxygenated sesquiterpenes found in the sample, however in much lower concentrations, about 3% each.

#### Infrared Analysis of the Essential Oil

Pure samples provided a spectrum typical of hydrocarbons with a certain degree of unsaturation (Figure 2). Absorptions at 2925 cm^−1^ and 2856 cm^−1^ refer to the Csp^3^-H stretching vibrations, and those at 1448 cm^−1^, 1382 cm^−1^ and 1357 cm^−1^ are Csp^3^-H bend vibrations. The very weak bands at 1633 cm^−1^ and just above 3000 cm^−1^ might be associated with Csp^2^-H vibrations, which are comparably less intense due to the low degree of unsaturation in the sample. The presence of C=C bonds is further evidenced by the intense absorption at 738 and 705 cm^−1^, referring to the =C-H out-of-plane bending.

### 2.2. Biological Assays of Essential Oil from V. tweediana

We evaluated the effects of *V. tweediana* essential oil on neutrophil migration in the air pouch model using carrageenan as a phlogistic agent. To assess the in vivo impact, we analyzed histological sections of air pouch tissue from vehicle-treated animals challenged with carrageenan (Figure 3A–C). These sections revealed typical morphological changes associated with inflammation, such as thickening of the air pouch membrane, intense edema, fibrin-hemorrhagic exudate, leukocyte accumulation, and tissue injury. Most of the inflammatory cells observed were neutrophils, identifiable by their lobulated nuclei. These cells were present in high numbers in the injured tissue and distributed throughout the adjacent muscular fibers.

Compared to mice injected with the vehicle, those treated with indomethacin displayed a reduction in the thickness of air pouch membranes challenged with carrageenan (Figure 3L–N). Furthermore, edema was restricted to specific regions, and there was a noticeable decrease in the number of neutrophils infiltrating the area. In addition to fibroblasts and neutrophils, few normal and foamy macrophages were present in the pouch membrane.

Animals treated with *V. tweediana*, particularly those treated with a dosage of 300 mg kg^−1^, displayed a noticeable reduction in the thickness of air pouches challenged with carrageenan (Figure 3I–K). The pouch wall was formed by loose connective tissue and newly formed blood vessels, and no edema was observed. Additionally, the number of neutrophils and macrophages were significantly decreased compared to the negative control group, and a lining of fibroblast-like cells with no directional orientation was observed. These histological observations suggest that *V. tweediana* essential oil effectively suppresses the acute inflammatory response in the pouch wall.

Figure 4A shows that the *V. tweediana* extract, administered at a dose of 300 mg kg^−1^, was able to reduce cell counts in the exudate similarly to indomethacin when compared to the control group. Furthermore, differential cell analysis indicated that both *V. tweediana* and indomethacin primarily affected neutrophils (Figure 4B).

In addition to reducing cell counts in the exudate, *V. tweediana* extract at a dose of 300 mg kg^−1^ also decreased the levels of TNF and IL-1β compared to carrageenan-injected mice, as shown in Figure 4C,D. Indomethacin-treated animals exhibited similar results. However, neither the essential oil nor indomethacin influenced IL-6 production, as demonstrated in Figure 4E.

## 3. Discussion

### 3.1. Essential Oil Extraction and Chemical Characterization

The dry leaves of *V. tweediana* when submitted to hydrodistillation provided essential oil in yield similar to other species of this genus. For instance, yields of 0.10 and 0.15% were observed for *V. brasiliana* [17] and *V. phosphorica* [18], respectively. On the other hand, *V. montevidensis* gave yields ranging from 0.19–0.21% for fresh leaves [4]. Nevertheless, these yields are considerably lower when compared to representatives of aromatic plants such as the genus *Piper* (Piperaceae), which can reach values of 0.96%, as observed for *P. mikanianum* [19].

The general methodology used to characterize the essential oil, which consisted of gas-phase chromatography with FID and MS detectors, was suitable and provided a level of characterization of 97.04%.

Terpenes were the main class of secondary metabolites present in the analyzed sample, with the predominance of hydrocarbon sesquiterpenes. Identical results were observed in the literature [17,18], except that the hydrocarbon monoterpenes β-pinene and myrcene were the major compounds in *V. montevidensis* [18] and in one work related to *V. phosphorica* [20], respectively. However, the major caryophyllenes (α and β), bicyclogermacrene, germacrene D, and spathulenol are common to this genus.

In contrast, the IR spectrum (Figure 2) does not allow for the identification of single compounds but instead provides a fingerprint of the mixture, which is useful as a preliminary analysis. Furthermore, it is a very accessible instrument, and the IR-ATR technique does not require any additional work during sample preparation. 

### 3.2. Biological Assays

To investigate the anti-inflammatory effects of *V. tweediana*, an in vivo experiment using the air pouch model induced by carrageenan was performed. Carrageenan is known to activate transcription factors such as STAT3 and NF-κB [21], leading to the production and secretion of various inflammatory mediators. The differential analysis of the air pouch-migrated cells, based on the total leukocyte count, showed that the reduction in leukocytes was mainly due to the decrease in polymorphonuclear cells, particularly neutrophil migration. Moreover, histological analyses of the pouch tissue revealed a significant reduction in leukocyte migration and connective tissue edema when compared to the carrageenan group.

During the inflammatory process, the production and release of inflammatory cytokines and chemokines, including TNF, IL-1β, and IL-6, are significantly increased [22]. The findings obtained in this study indicate that treatment with *V. tweediana* extract led to a significant decrease in TNF and IL-1β levels, but no effect on IL-6 was observed. TNF and IL-1β are pro-inflammatory cytokines that play a pivotal role in the initiation and progression of inflammation. They can induce the production and secretion of other cytokines and chemokines, the expression of adhesion molecules, and angiogenesis and mediate systemic inflammatory effects, such as fever, hypotension, and the promotion of neutrophilia. The reduction of TNF and IL-1β levels directly implies a decrease in the progression of the inflammatory process. This reduction results in decreased cell migration and production/secretion of inflammatory cytokines, which in turn contributes to the restoration of homeostasis [23]. IL-6 is a cytokine that is not only released by neutrophils but also by other cell types such as keratinocytes, fibroblasts, and endothelial cells. This broad release by multiple cell types justifies its increased levels in the in vivo model [24]. IL-6 is generally considered a pro-inflammatory cytokine; however, there is some evidence indicating that under certain conditions, it possesses anti-inflammatory properties [25].

The effect observed could be correlated with the presence of β-caryophyllene, which is a bicyclic sesquiterpene that has garnered Food and Drug Administration approval for its promising therapeutic potential. β-caryophyllene binds to cannabinoid receptor type 2 (CB2), primarily expressed in immune and immune-derived cells, which modulate inflammatory responses. Studies on arthritic rats showed that β-caryophyllene possesses analgesic, anti-inflammatory, and antioxidant properties. Recent research suggests that β-caryophyllene’s interaction with peroxisome proliferator-activated receptor gamma (PPARγ) may provide some explanation [26].

Furthermore, the essential oil studied herein presents other compounds that may be responsible for its anti-inflammatory activity, such as α-caryophyllene [27], spathulenol [28], and germacrene [29].

Finally, the use of *V. tweediana* as a phytoterapic is consistent with the present findings and might be related to an agonist synergism involving volatiles and non-volatile metabolites [10].

## 4. Materials and Methods

### 4.1. Plant Material

The collection of leaves was carried out on 12 October 2019 on the banks of Rua Presidente Costa e Silva, nº 22, Testo Rega, Pomerode, SC (26°42′42.5″ South Latitude and 49°09′51.1″ West Longitude). The intact leaves were placed to dry on a bench at room temperature and in the shade, remaining there until constant mass. Exsiccates of *V. tweediana* were deposited in the Dr. Roberto Miguel Klein herbarium under the code FURB63250 under the supervision of the curator prof. Dr. André Luís de Gasper.

### 4.2. Essential Oil Extraction

After drying, the leaves were cut into pieces and subjected to hydrodistillation in a modified Clevenger apparatus for 4 h. A proportion of 50 g of plant material to 1 L of deionized water was used. To isolate the EO, the liquid-liquid extraction technique was used with bidistilled dichloromethane as the extracting solvent. The essential oil and hydrolate were transferred to a separation funnel and subjected to 3 (three) successive extractions. The organic fractions were combined in an Erlenmeyer flask, with portions of desiccant agent (magnesium sulfate) added to the flask. The mixture was then filtered into a round-bottom flask and concentrated in a rotary evaporator. The vial containing the EO was kept in a desiccator under vacuum and subsequently weighed to ensure the complete removal of the solvent. The percentage extraction yield was the quotient of the mass of the isolated oil by the mass of the plant material used in the extraction times 100 (one hundred). The oil was stored in a vial and kept refrigerated until analysis.

### 4.3. Chemical Characterization of the Essential Oil

For the quantification of the EO constituents, a semi-quantitative method was used based on their peak areas without correcting response factors. The concentrations are the mean values of the two analyses. This was carried out using a GC-FID, using an instrument from the Perkin-Elmer brand, model Clarus 600. For the identification of its constituents, we used a Perkin-Elmer gas chromatograph, model Clarus 680 coupled to a Perkin-Elmer model 600T mass detector.

The sample was diluted in an appropriate vial with double-distilled dichloromethane at a concentration of approximately 1% (m/m). The total analysis time was 62 min for both detectors. The injection volume of the samples was 1 microliter in split mode 1:20, with helium as the carrier gas in a flow of 1 mL min^−^^1^ and DB5 type column measuring 30 m (length), 0.25 mm (internal diameter) and 0.25 μm (thickness of stationary phase). The instruments operated in temperature gradient mode according to Adams [16]: initial column temperature of 60 °C, with a heating rate of 3 °C min^−1^ reaching 240 °C, remaining at that temperature for 2 min. In the GC-FID analysis, the injector temperature was 220 °C and 280 °C for the detector. In the GC-MS analysis, the temperature of the ion source and the transfer line were 250 °C and 240 °C, respectively, and the instrument operated in electron impact mode, with a collision energy of 70 eV, under the same analytical conditions described for GC-FID.

The arithmetic retention index (AI) of each constituent present in the sample was experimentally determined in GC-FID using a homologous series of linear alkanes and applying the Van den Dool and Kratz Equation (1) [16]:AI = 100*Z* + 100 (*R*T_*X*_ − *R*T_*Z*_/*R*T_(*Z*+1)_ − *R*T_*Z*_)(1)
where Z: alkane with Z carbons; (Z + 1): alkane with Z + 1 carbons; RT_Z_: retention time of the alkane with Z carbons; TR_Z+1_: retention time of the alkane with Z + 1 carbons; and TR_X_: retention time of the constituent x.

Identification of the constituents was carried out using an automatic interpretive program of the NIST spectra library (2008) by visual comparison of the spectra obtained and those available in the literature and by comparing the experimental AI for each of the constituents and those from the literature [16].

Finally, to establish the spectroscopic profile of the essential oil and the identification of its main functional groups, infrared spectroscopy with attenuated total reflectance (IR-ATR) was conducted using Bruker equipment, model Vertex 70.

### 4.4. Animals 

Experiments were performed in male Swiss mice, 6–8 weeks old, (n = 6). All animals were obtained from the Central Animal Facility of the Universidade Regional de Blumenau (FURB) and kept in a climate-controlled room at 22 ± 2°C, under a light/dark (12:12 h) cycle, with water and food ad libitum.

### 4.5. Leukocyte Migration: Air Pouch Model 

Air pouches were produced on the back of the mice. Briefly, the animals were orally treated by gavage with *V. tweediana* essential oil at doses of 50, 150, or 300 mg kg^−1^, indomethacin (30 mg kg^−1^, positive control), or vehicle (PBS; 10 mL kg^−1^). After 1 h, carrageenan (1%; 3 mL/cavity) was injected directly into the air pouch chamber. Four hours later, a small incision was made in the air pouch, the cavity was washed with 3 mL of PBS, and the inflammatory infiltrate was collected for analysis. Total and differential cell counts were performed, and the rest of the exudate was kept for further analysis of cytokines.

### 4.6. Histological Analysis

The air pouch lining tissue was collected for histological analysis and preserved in formaldehyde solution until the blades were made. The preparation consisted of inserting the tissue fragment into a cassette. The sample was submitted to a 1 h dehydration process in increasing concentrations of ethanol (70%, 96%, and absolute ethanol). Subsequently, each sample was placed in xylol and then in a paraffin bath. After this process, the cassette with the sample was placed in a microtome where serial sections of 3 or 4 µm were made. The sections were placed on slides and stained with hematoxylin and eosin. The samples were then analyzed via optical microscopy (4, 10 and 40×).

### 4.7. Cytokines Quantification

The levels of tumor necrosis factor (TNF), interleukin (IL)-6, and IL-1β were analyzed in the exudate samples from the air pouch model. The assays were performed according to the manufacturer’s instructions (DuoSet R&D Systems—Minneapolis, MN, USA), and the results are expressed in pg mL^−1^.

### 4.8. Statistical Analysis

The obtained results are expressed as the mean ± standard error of the mean (SEM) and were statistically analyzed using analysis of variance with multiple comparisons (ANOVA), and, when necessary, the Tukey–Kramer or Dunnett’s post hoc test was used. The data were analyzed using the GraphPadPrism^®^ program, assuming *p* < 0.05 as statistically significant.

## 5. Conclusions

The essential oil from the leaves of *V. tweediana* was characterized by means of GC-FID, GC-MS, and IR-ATR techniques, and hydrocarbon sesquiterpenes make up the largest group present. Biological assays demonstrated notable anti-inflammatory and immunomodulatory activities, including reduced leukocyte migration and modulated cytokine secretion (IL-1β, IL-6, and TNF). These effects are likely linked to the high caryophyllene content. The research suggests the necessity of further studies to assess the synergism between the oil’s constituents and to determine the influence of seasonality and circadian rhythm on its chemical composition and production.

The current work is another example of the importance of plant biodiversity, its study, and preservation.

## Figures and Tables

**Figure 1 pharmaceuticals-17-01492-f001:**
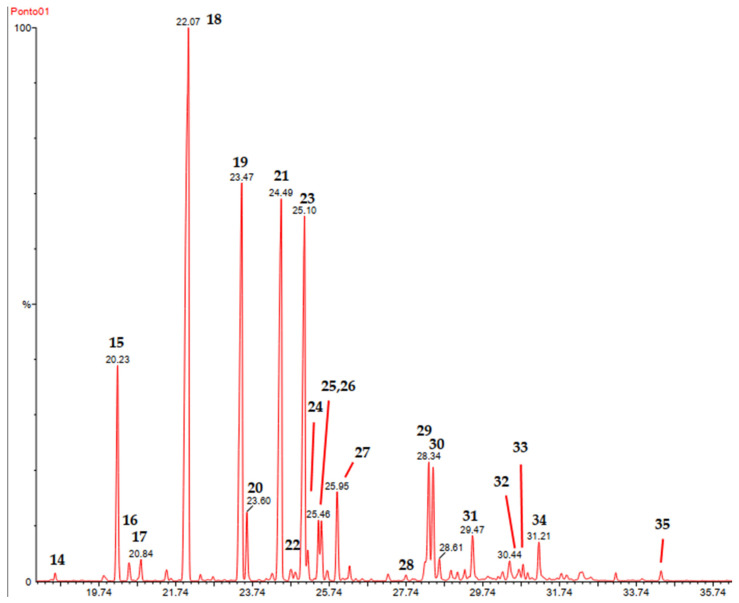
Expanded GC-MS chromatogram of the essential oil from the dry leaves of *V. tweediana*. In the selected region, 18–36 min are the major constituents of the essential oil. Peaks are assigned with entries 14–34.

**Figure 2 pharmaceuticals-17-01492-f002:**
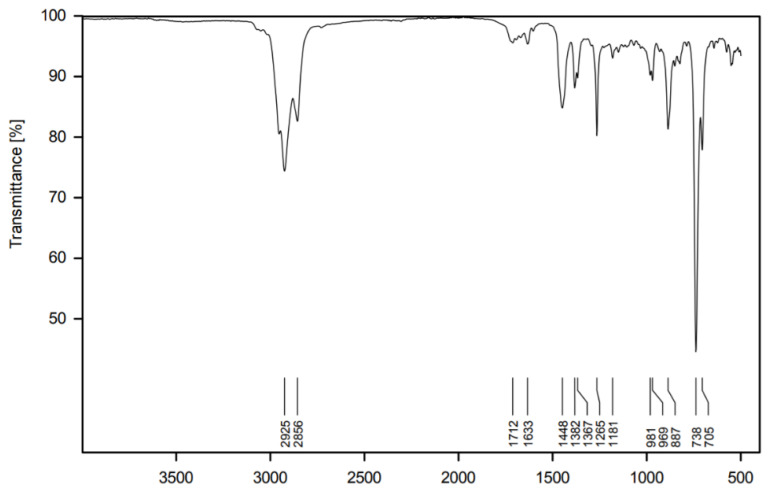
Infrared spectrum of the pure essential oil of *V. tweediana*.

**Figure 3 pharmaceuticals-17-01492-f003:**
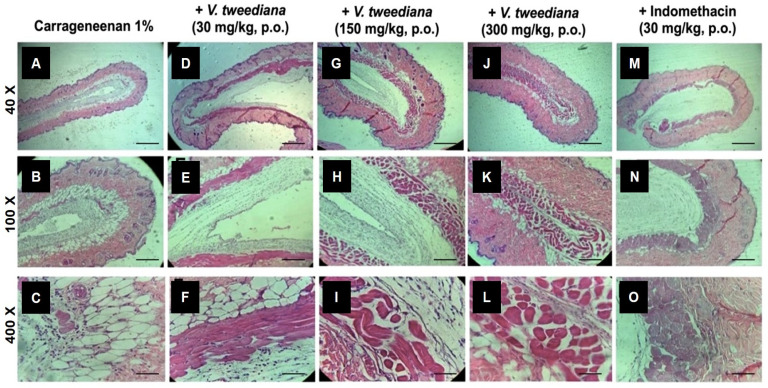
Air pouches were induced in the dorsal subcutaneous tissue of Swiss mice. Illustrative figures of air pouches tissues obtained from animals treated orally with (**A**–**C**) vehicle, (**D**–**F**) *V. tweediana* 30 mg/kg, (**G**–**I**) *V. tweediana* 150 mg/kg, (**J**–**L**) *V. tweediana* 300 mg/kg, (**M**–**O**) Indomethacin 30 mg/kg one hour before the injection of 2 mL of carrageenan (1%). Representative H.E. histological sections of skin biopsies obtained from mice with air pouches (40, 100, and 400×). Scale bar = 100 μm.

**Figure 4 pharmaceuticals-17-01492-f004:**
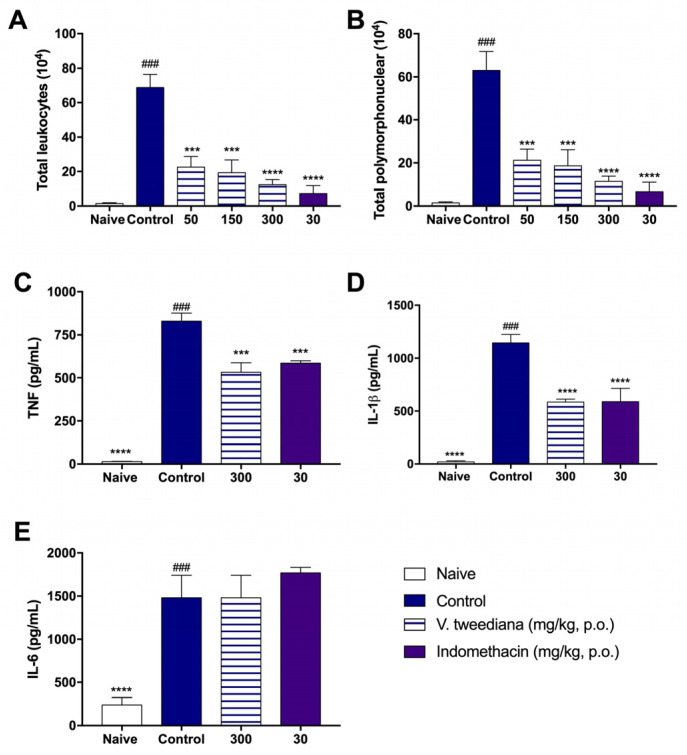
An air pouch was induced in the dorsal subcutaneous tissue of Swiss mice. The animals were treated orally one hour before the injection of 2 mL of carrageenan (1%). The lavage of the inflammatory infiltrate was collected 4 h after the injection of carrageenan into the air pouch. The determination of the (**A**) total number of exudate cells was performed in a Neubauer chamber and the (**B**) differential value was determined. The levels of (**C**) TNF, (**D**) IL-1β, and (**E**) IL-6 were determined using the ELISA method. Values are expressed as the mean ± S. E. M. of tests performed with the inflammatory exudate obtained from 6 animals per group. *** *p* < 0.001 and **** *p* < 0.0001 vs. the control group significantly different from the naïve group ### *p* < 0.0001 (one-way ANOVA followed by Tukey’s post hoc test).

**Table 1 pharmaceuticals-17-01492-t001:** Chemical composition of the essential oils from the dry leaves of *V. tweediana* (Baker) H. Rob. (Asteraceae).

Entry	Constituents	AI_exp_	AI_lit_	[%] SD
1	α-pinene	935	932	0.27 ± 0.01
2	benzaldehyde	974	952	0.15 ± 0.01
3	β-pinene	980	974	0.58 ± 0.01
4	β-myrcene	991	988	0.27 ± 0.00
5	α-phellandrene	1009	1002	0.30 ± 0.01
6	cymene ^1^	1026	1020 ^2^/1022 ^3^	0.39 ± 0.01
7	limonene	1030	1024	0.37 ± 0.01
8	*Z*−β−ocimene	1035	1032	0.67 ± 0.01
9	*E*- β-ocimene	1046	1044	0.22 ± 0.01
10	γ-terpinene	1059	1054	0.09 ± 0.00
11	terpinen-4-ol	1188	1174	0.16 ± 0.01
12	β-cyclocitral	1229	1217	0.09 ± 0.00
13	safrole	1298	1285	0.08 ± 0.00
14	δ-elemene	1334	1335	0.64 ± 0.00
15	α-copaene	1373	1374	5.07 ± 0.01
16	β-bourbonene	1380	1387	0.37 ± 0.01
17	β-elemene	1388	1389	0.59 ± 0.00
18	β-caryophyllene	1419	1417	27.73 ± 0.02
19	α-caryophyllene	1454	1452	15.12 ± 0.03
20	*allo*-aromadendrene	1457	1458	1.57 ± 0.01
21	germacrene D	1479	1480	13.98 ± 0.01
22	*E*-β-ionone	1487	1487	0.26 ± 0.02
23	bicyclogermacrene	1495	1500	13.32 ± 0.00
24	α-muurolene	1497	1500	2.56 ± 0.00
25	*E*,*E*−α−farnesene	1505	1505	0.28 ± 0.01
26	β-bisabolene	1507	1505	0.18 ± 0.00
27	δ-cadinene	1519	1522	1.99 ± 0.03
28	*E*-nerolidol	1565	1561	0.34 ± 0.01
29	spathulenol	1580	1577	2.94 ± 0.00
30	caryophyllene oxide	1583	1582	3.24 ± 0.00
31	humulene-1,2-epoxide	1611	1608	1.07 ± 0.04
32	τ-muurolol	1647	1640	0.48 ± 0.02
33	α-muurolol	1651	1644	0.48 ± 0.00
34	α-cadinol	1658	1652	0.78 ± 0.01
35	6*R*,7*R*-bisabolone	1748	1740	0.21 ± 0.01
36	kaurene ^4^	-	2042	0.20 ± 0.01
	Total			94.07
	HM			3.16
	OM			0.25
	HS			83.40
	OS			9.80
	HD			0.20
	Others			0.23

^1^ It was not possible to discriminate between the two isomers: ^2^
*para* and ^3^
*ortho*; AI_exp_: experimental arithmetic retention indices; AI_lit_: arithmetic retention indices from the literature [16]; ^4^ Based on MS analysis; HM: hydrocarbon monoterpenes; OM: oxygenated monoterpenes; HS: hydrocarbon sesquiterpenes; OS: oxygenated sesquiterpenes; HD: hydrocarbon diterpene.

## Data Availability

The original contributions presented in this study are included in the article/Appendix A. Further inquiries can be directed to the corresponding author(s).

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
