# Peer review of "Vernonanthura tweediana (Baker) H. Rob. (Asteraceae), an Ordinary Bush or an Anti-Inflammatory and Immunomodulator Aromatic Species?"

_pharmaceuticals, 2024, doi:10.3390/ph17111492_

Round 1

Reviewer 1 Report

Comments and Suggestions for Authors

The aim of this paper is to characterize the chemical composition and biological properties of the essential oil of V. tweediana. The essential oil is characterised as a whole, without distillation and characterisation of different fractions. This will be done in further research.

Comments :

Line 86. Please correct the sentence and write: “The sample was analysed by gas chromatography using a flame ionization detector (GC-FID) and a mass detector (GC-MS)”. Figure 1 shows the chromatogram obtained by GC-FID for the essential oil.

Line 88. The authors should refer about “semi-quantitative” analysis because the percentage of the compounds is obtained from the percentage of the areas. No calibration standards are used for quantification.

The authors should also explain why the “semi-quantitative” analysis is done from GC-FID data and not from GC-MS data. Theoretically, the essential oil was analysed with GC-FID and GC-MS.

In Figure 1 the chromatographic peaks should be identified. The authors might consider zooming in on the interesting part of the chromatogram and showing the identified and unidentified compounds.

The authors state that identification is based on retention index and MS. The % similarity of the MS spectrum for each identified compound could be shown in Table 1. It might be more useful to give the retention time instead of the entry. It might also be good to give a measure of uncertainty. At least, the analysis could be performed three times and the mean value together with the standard deviation of the three values could be given.

The authors might consider giving the GC-MS chromatogram together with the MS spectra of the compounds and the MS spectra of the identified compound from the library. This could be given in the supporting information.

In my opinion, IR spectra are not very informative in the case of a mixture of compounds. It only allows you to say that the essential oil is mainly unsaturated. This information is not very relevant.

Lines 259-262. The authors should correct the symbol for degrees Celsius. It is correct for the temperature of 220, but not for the other temperatures.

I think the authors could include some conclusions after the discussion of the results.

I would put the materials and methods after the introduction. That way it is easier to follow how the results were obtained.

Author Response

Dear Reviewer #1. We thank you for handling our manuscript. All your comments are answered below, and the suggested modification improved the quality of the manuscript. All modifications are highlighted in red in the revised version of the manuscript.

Comment 1: Line 86. Please correct the sentence and write: “The sample was analysed by gas chromatography using a flame ionization detector (GC-FID) and a mass detector (GC-MS)”. Figure 1 shows the chromatogram obtained by GC-FID for the essential oil.

Response 1: Thank tou for pointing this out. We have corrected as suggested (see page 2, lines 93/94);

Comment 2: Line 88. The authors should refer about “semi-quantitative” analysis because the percentage of the compounds is obtained from the percentage of the areas. No calibration standards are used for quantification.

Response 2: We agree with your comments. This information is given in page 9, lines 276/277 (heading Materials and Methods).

Comment 3: The authors should also explain why the “semi-quantitative” analysis is done from GC-FID data and not from GC-MS data. Theoretically, the essential oil was analysed with GC-FID and GC-MS.

Response 3: This information was added in details to the manuscript. See page 2, lines 94-97 and page 3, lines 98-102.

Comment 4: In Figure 1 the chromatographic peaks should be identified. The authors might consider zooming in on the interesting part of the chromatogram and showing the identified and unidentified compounds.

Response 4: The suggestion was taken into consideration. The GC-FID chromatogram was substituted by GC-MS chromatogram (expanded in the region 18-36 min) with a better resolution and peaks are now related to the entries, according to Table 1. See page 3, Figure 1. The text related to the chromatogram and Table 1 has also been improved. See page 3, lines 103-106; page 3, lines 117-120 and page 4, lines 127-131.

Comment 5: The authors state that identification is based on retention index and MS. The % similarity of the MS spectrum for each identified compound could be shown in Table 1. It might be more useful to give the retention time instead of the entry. It might also be good to give a measure of uncertainty. At least, the analysis could be performed three times and the mean value together with the standard deviation of the three values could be given.

Response 5: We partially agree with the former comment. The entries are now related to the peaks. See Figure 1 and Figure 3SA-C (suplementary material). Concerning the analysis, is now informed that it was performed twice and that the concentration is a mean value together with the standard deviation. See Table 1 and the heading "Materials and Methods", page 9, lines 277-278.

Currently, all samples of our lab are analysed three times, but in the past if a good level of reproductibility was obtained, a third analysis would not be conducted. A comment is made regarding the standard deviation observed. See page 3, lines 101/102.

Comment 6: The authors might consider giving the GC-MS chromatogram together with the MS spectra of the compounds and the MS spectra of the identified compound from the library. This could be given in the supporting information.

Response 6: We have done as suggested. Supporting Material is now provided with all the mass spectra of the identified compounds and hits from NIST library and when appropriate the mass spectrum from literature [17]. Furthermore, 3 expanded chromatograms are provided with 36 entries, corresponding to all the identified compounds (Figure 3S A-C). GC-FID chromatogram with improved quality is also provided (Figure 1S).

Comment 7: In my opinion, IR spectra are not very informative in the case of a mixture of compounds. It only allows you to say that the essential oil is mainly unsaturated. This information is not very relevant.

Response 7: A short text has been added to the IR analysis in the heading “Discussion” (see page 8, lines 210-213). We would like to pointing it out that IR also indicates the absence of alcohols, amines, ethers, carbonyls and other functions, or they are present in very low concentrations. Is is useful for a preliminar analysis, but also it gives a fingerprint of the sample.

Comment 8: Lines 259-262. The authors should correct the symbol for degrees Celsius. It is correct for the temperature of 220, but not for the other temperatures.

Response 8: Thank you for the observation. It was corrected as suggested. See page 9, lines 287-290.

Comment 9: I think the authors could include some conclusions after the discussion of the results.

Response 9: Thank you for your observation. Please note that we included some conclusions that support our manuscript. See heading 5, page 10, lines 338-348.

Comment 10: I would put the materials and methods after the introduction. That way it is easier to follow how the results were obtained.

Response 10: Although your observation is important, we have followed the template provided by the editor.

Reviewer 2 Report

Comments and Suggestions for Authors

attached

Comments on the Quality of English Language

There are many awkward phrases and sentences throughout the manuscript. Please proofread the the manuscript with someone well-versed in English.

Author Response

Dear Reviewer #2. We thank you for handling our manuscript. All your comments are answered below and the suggested modification improved the quality of the manuscript. All modifications are highlighted in red in the revised version of the manuscript.

Comment 1: There are many awkward phrases and sentences throughout the manuscript. Please proofread the the manuscript with someone well-versed in English.

Response 1: Thank you for pointing this out. The English was revised, and all modifications were highlighted in red.

Comment 2: While the study identifies key compounds, it doesn't isolate and test these individually, limiting the ability to pinpoint which specific components are responsible for the biological activity. 

Response 2: The present manuscript focuses on the chemical characterization of Vernonanthura tweediana essential oil and its synergistic effects on the immune response.  At this moment it is not feasible the fractionation of the essential oil or any attempt to isolate some of its constituents. But due to its relevance, it is proposed as future work. See heading "Conclusions", page 10, lines 338-348.  

Comment 3: The yield of 0.13% might be too low for practical applications, which could affect its feasibility as a potential therapeutic product. 

Response 3: A yield of 0.13%, while low, is within the normal range for many essential oils, as yields can vary significantly depending on the plant species, extraction methods used, and other factors associated to seasonality and circadian rhythm. Further study is necessary to enhance scalability. This has been mentioned in the heading “Conclusions”.  

Comment 4: According to the study's findings, the essential oil yield from the dried leaves is only 0.13%, and the effective/bioactive dose was 300 mg/kg of body weight. This means that if we extract essential oil for the treatment in humans, for a normal 70 kg human, we would need 300 mg X 70 = 21000 mg or 21 g of essential oil in one dose. We would  21 grams of essential oil (for one dose for a 70 kg body weight of a person), given a 0.13% yield. This looks pretty impractical. 

Response 4: To translate a dose from mice to humans, researchers typically use the body surface area (BSA) conversion method. This approach accounts for differences in metabolism and physiology between species. The most common method involves converting the dose based on body weight using a conversion factor known as the human equivalent dose (HED). In this context, the human equivalent dose (HED) based on a dose of 300 mg/kg in mice is approximately 24.32 mg/kg. Additionally, is important to mention that a lower dose in mice (50 mg/kg) present significant biological effects, which reduces the equivalent dose in humans.

Comment 5: The study doesn’t address the essential oil's toxicity or potential side effects, an important factor when proposing therapeutic applications.

Response 5: Although this is true, it must be considered the primary focus of this manuscript which is to demonstrate the biological effects of Vernonanthura tweediana and to characterize its essential oil. We believe this represents a significant step forward. Certainly, we recognize that a comprehensive toxicological evaluation will be necessary in the future to fully assess its safety for therapeutic applications.

Comment 6: Only the short-term effects of essential oils have been studied. The study lacks information on the long-term effects and possible side effects. 

Response 6: While it is true that this study focuses on short-term effects, this approach was intentional as it utilizes an acute model of inflammation, specifically the air pouch model. This model is particularly effective for assessing immediate biological responses and is widely recognized for its ability to mimic acute inflammatory conditions. By concentrating on short-term effects, we aimed to establish a foundational understanding of the essential oil's therapeutic potential in acute scenarios.

Comment 7: The expression of only selected markers has been studied. It would have been more informative if an unbiased approach like transcriptomic, proteomic, or metabolomic studies were conducted to see the effect of oil on overall gene expression or metabolic pathways. 

Response 7: While acknowledging that this study focused on selected inflammatory markers, this approach was justified given the specific aims of the research. It was measured by key inflammatory mediators that are critical in understanding the immediate biological effects of the essential oil on inflammation. These selected markers provide valuable insights into the oil's anti-inflammatory properties and its mechanisms of action.

Additionally, it was conducted histological analyses to complement the biochemical data. This allowed the visualization of the effects of the essential oil on tissue architecture and inflammation, providing a more comprehensive understanding of its impact at the cellular level.

While transcriptomic, proteomic, or metabolomic approaches would certainly offer a broader perspective on gene expression and metabolic pathways, they also come with increased complexity and resource requirements. The selected targeted approach prioritizes the identification of specific pathways and mechanisms, laying a solid foundation for future studies that could explore these broader methodologies. This initial focus is essential for developing a deeper understanding of the essential oil's therapeutic potential before expanding to more extensive, unbiased analyses.

Comment 8: Though the effect of essential oil treatment on the expression of TNF and IL genes has been studied, the specific mechanism through which the oil can exert its impact has not been explored.

Response 8: It could be a topic for future research.  

Comment 9: Lines 4 and 26, the word "specie" should be "species," as "species" is both singular and plural.

Response 9: Than you for this observation. They have been corrected. See title (line 4) and abstract (line 26).

Comment 10: Line 24, “Oral traditional knowledge disseminate.” Since the subject is singular, it should be written as “Oral traditional knowledge disseminates.”

Response 10: It was corrected (line 24).

Comment 11: Line 24, “Oral traditional knowledge disseminate.” Since the subject is singular, it should be written as “Oral traditional knowledge disseminates.”

Response 11: It was corrected (line 24).

Comment 12: Rephrase the sentence starting from “Among them” in lines 44 to 48.

Response 12: It was done as suggested (lines 44-46).

Comment 13: Line 58, replace “The treatment of inflammatory processes” with “The treatment of inflammatory conditions or diseases.”

Response 13: It was done as suggested (line 58).

Comment 14: Rephrase the sentence in lines 72 to 75 for clarity.

Response 14: It has been replaced by the following sentence: "Biological assays demonstrated notable anti-inflammatory and immunomodulatory activities, including reduced leukocyte migration and modulated cytokine secretion. These effects are likely linked to the high caryophyllene content". Page 2, lines 80-82.

Comment 15: The quality/resolution of Figure 1 is not very good. Replace it with a high-resolution image.

Response 15: It has been replaced and additional information was introduced (peaks associated with entries).

Comment 16: Line 101, change “It was not possible discriminate between” to “It was not possible to discriminate between.”

Response 16: It was corrected (page 4, line 123).

Comment 17: Write all species names in italic format.

Response 17: Thank you for the observation. It was made as requested.

Round 2

Reviewer 1 Report

Comments and Suggestions for Authors

The authors have modified the manuscript according to my suggestions. The manuscript is suitable for publication.

Reviewer 2 Report

Comments and Suggestions for Authors

Thank you for addressing all of my concerns. I have no additional comments.